# Evaluating cancer etiology and risk with a mathematical model of tumor evolution

Sophie Pénisson [1,2,3,4,5], Amaury Lambert[6,7] & Cristian Tomasetti [1,2,3] ✉

Recent evidence arising from DNA sequencing of healthy human tissues has clearly indicated that our organs accumulate a relevant number of somatic mutations due to normal endogenous mutational processes, in addition to those caused by environmental factors. A deeper understanding of the evolution of this endogenous mutational load is critical for understanding what causes cancer. Here we present a mathematical model of tumor evolution that is able to predict the expected number of endogenous somatic mutations present in various tissue types of a patient at a given age. These predictions are then compared to those observed in patients. We also obtain an improved fitting of the variation in cancer incidence across cancer types, showing that the endogenous mutational processes can explain 4/5 of the variation in cancer risk. Overall, these results offer key insights into cancer etiology, by providing further evidence for the major role these endogenous processes play in cancer.

The mathematical modeling, and associated statistical analysis, of the evolution of mutations in cell populations and bacteria has a relatively long history. When combined with epidemiological or experimental data it has yielded several important findings. For example, the seminal work of Luria and Delbrück[1], demonstrating the critical role played by randomly occurring mutations arising before—rather than after—the beginning of treatment as the cause behind the development of resistance to antibiotics. Their Nobel Prize winning work led to a flurry of further modeling work to understand the development of resistance to chemotherapy and HIV antiretroviral treatment, highlighting the importance of combination therapy. Another example is provided by the models and analyses produced by Charles and Luce-Clausen[2], Nordling[3], and Armitage and Doll[4], which shed light on the multi-stage nature of cancer evolution.

In the last few years, a series of papers based on mathematical models and statistical analyses, in combination with epidemiological, sequencing, and other experimental data, have provided evidence for the first time of the large role in tumor evolution played by the normal endogenous processes occurring in human tissues[5–11]. In refs. 6 and 7 a highly significant, strong correlation (Spearman's rho = 0.81, $P < 3.5 \times 10^{-8}$) between the total number of stem cell divisions in an organ and the lifetime risk in that organ was reported. No known common environmental exposure or inherited factor is present among the many cancer types analyzed. Specifically, no common exposure—beside aging—was shared by more than five cancer types, out of the 25 analyzed (27 with the inclusion of breast and prostate cancer in ref. 7). And generally exposures were shared by only 2–3 tissues at most, with many cancer types having no known exposures increasing their risk. Given the large number of cancer types analyzed, from a statistical point of view this then led to the reasonable assumption that the effects of lifestyles and environmental (E) exposures and inherited (H) factors on the overall correlation discovered were greatly reduced, by essentially averaging them out (see ref. 7 for more details). The most plausible conclusion was then that endogenous random mutational processes naturally occurring in human tissues could explain a large fraction—estimated to be equal to 66%—of the variation in cancer risk

[1]Center for Cancer Prevention and Early Detection, City of Hope, Duarte, CA 91010, USA. [2]Division of Mathematics for Cancer Evolution and Early Detection, Department of Computational & Quantitative Medicine, Beckman Research Institute, City of Hope, Duarte, CA 91010, USA. [3]Division of Integrated Cancer Genomics, Translational Genomics Research Institute, Phoenix, AZ 85004, USA. [4]Univ Paris Est Creteil, CNRS, LAMA, F-94010 Creteil, France. [5]Univ Gustave Eiffel, LAMA, F-77447 Marne-la-Vallée, France. [6]SMILE Group, Center for Interdisciplinary Research in Biology, Collège de France, 75231 Paris, Cedex 05, France. [7]Institut de Biologie de l'École Normale Supérieure (IBENS), CNRS UMR8197, INSERM U1024, 75230 Paris, Cedex 05, France. ✉e-mail: ctomasetti@coh.org

across tissues and that they represent an important source of cancer. We termed this endogenous source of mutations "bad luck" or $R$. Importantly, these findings have been since then supported by several further analyses, conducted by us and others, using a variety of different methodologies, and confirming the presence of a large number of mutations attributable to $R$ factors, with a mutational contribution that can be currently estimated to be around two-thirds of all mutations found in cancers, and certainly not smaller than 40%[7–17].

To make further progress in our understanding of tumor evolution and cancer etiology, a more complete mathematical simulation model was recently presented in ref. 10, where all phases of a tissue history were included, from tissue development to homeostasis, and all the way to cancer occurrence. In that model, the different types of cell division and the fitness effects of different types of driver mutations were accounted for. The model also included a carrying capacity to prevent the unrealistic assumption of unlimited exponential growth, typical in the mathematical modeling of tumor evolution. That model was able to reproduce qualitatively the incidence of multiple cancer types. This suggested that many of the basic ingredients of tumor evolution are common across different tissue types and can be captured by a mechanistic model, when accounting for the fundamental differences among those tissues, like the total number of cells and the cell division rate. The model provided several insights into both the timing of tumor evolution and the order of its driver mutational events. It also supported the idea that the background mutation rate plays a major role in cancer causation, being able to explain a substantial fraction of cancer incidence. A limitation of that work, however, was that it was based on simulations due to model complexity.

In the present work, we introduce a mathematical model of tumor evolution that is able to incorporate into analytical form the key ingredients of that simulation model[10] and further expand upon it. This represents important progress in the field of mathematical modeling of cancer evolution as closed-form analytical formulas enable a simpler, faster, and especially deeper understanding of the role played by each of the parameters included in the model. We also show how the time to cancer can be approximated by the Weibull distribution, providing a justification for this distribution which has been used for decades in survival analysis of cancer data but without a solid foundation. Most importantly, the model is used to predict the expected number of mutations that should be present in a cancer if only the endogenous mutational processes ($R$) were to be present and compare these predictions with the actual values observed via sequencing studies in various tissues. These analyses provide insights into cancer etiology by resulting in an overall fitting of the mutational load observed in cancer patients, with the exception of a few cancer types in which well-known exposures are present. We also use the model to reassess the original analysis of the role played by $R$ factors in cancer as presented in ref. 6.

## Results
### Predicted versus observed number of mutations
We can estimate the number of somatic mutations found in a cell of a healthy individual of age $a$ in the absence of $E$ and $H$ factors as

$$\eta^h(a) = \mu(D_0 + ba), \tag{1}$$

where $\mu$ is the somatic mutation rate (expected number of somatic mutations per cell division), $D_0$ the number of divisions before birth, and $b$ the cell division rate. In contrast, for a cancer patient of the same age, this expected number of mutations is

$$\eta^{exp}(a) = \mu\left(D_0 + \frac{b + b^{(1)} + \ldots + b^{(n-1)}}{n} a_n^*\right). \tag{2}$$

The difference between formula (1) and formula (2) is that instead of a constant cell division rate $b$, we now have the average of division rates $b^{(i)}$ that keep increasing the more driver mutations hits are present in that cell. The other difference between the two formulas is that instead of the patient's age $a$ we now have the expected age $a_n^*$ at which the first cancer cell with the $n$ required driver mutation hits has arrived (see mathematical modeling in the Methods section for its estimation). Using formula (2) we can compare our prediction for the total number of somatic mutations found when sequencing the cancer type of a patient of age $a$ with the actual number of mutations observed via the bulk sequencing of that individual cancer, using the public genomic TCGA database. The comparison is provided in Fig. 1a for 10 cancer types. We obtain a strongly significant correlation between our predictions and the actual observed values (Spearman's rho = 0.56, $P < 2.2 \times 10^{-16}$). In Fig. 1a, the placement of the points with respect to the identity line $y = x$ is naturally very sensitive to the choice of the expected number of mutations across the genome per cell division, which we set equal to 3, a typical value found in the literature (resulting in $\mu = 0.03$ on the exome, see ref. 5). However, no matter what the background mutation rate $\mu$ is chosen to be, the strong correlation we found between our predictions and the actual observations would stay the same. If we now add two more cancer types—lung (LUAD) and melanoma (SKCM)—we obtain Fig. 1b, with a correlation that drops to 0.26, which is not surprising since we only accounted for the endogenous mutational processes ($R$) in our predictions, and not for the extra mutational load that smoking and UV light exposures respectively add to them. In Fig. 1b, there is a large variation observed within each cancer type. Partially, this is because our estimation does not include the effects of environmental or inherited factors and the difference in exposure to those factors among patients with the same cancer type. However, an important observation is that the clear outliers in that figure are melanoma and lung cancer, the two cancer types whose etiology is most strongly affected by known $E$ factors. Even if today we still had not discovered the harmful effect of tobacco smoking and excessive UV light exposure through epidemiological studies, our predictions in Fig. 1b would provide evidence for the existence and major contribution of $E$ or $H$ factors. It is then critical to realize that the absence in Fig. 1b of major outliers beyond those caused by UV light and tobacco smoking, at least at the population level, suggests that no other factors with comparable large mutational effects should be present in the cancer types considered, even if still unknown to us today. Indeed, the genome would record their harmful mutational effects anyway.

In contrast, we obtain Fig. 1c when we include the effects of tobacco smoking and UV light, by using for both lung and skin cancer a four-fold increase of the average number of mutations per cell division on the exome. This rough estimation for the increase of the mutation rate is based on: 1) ref. 18 for smoking, and 2) the average ratio of 3.6 obtained in ref. 12 between the yearly number of mutations in sun-exposed versus sun-shielded skin, an underestimation of the increase due to UV light exposure since UV-induced mutations are also prominent in sun-shielded sites[19]. The inclusion of even just these two environmental factors yields an improvement in the fitting of our predictions to the observed values, with a correlation going from 0.26 in Fig. 1b to 0.67 in Fig. 1c.

We provide a summary of the parameters estimates used in our model, as well as the quantitative results obtained from it in Table 1. We computed for each cancer type the expected number of mutations in healthy individuals versus cancer patients in the absence of $E$ and $H$ factors thanks to formulas (1) and (2), where the expected value is calculated at the median age of the patients in the TCGA database. We provide also the interquartile range for the total number of mutations observed among those patients. We would like to point out that in these predictions there is no fitting of any of the parameters to the observed sequencing data. The number of divisions during the tissue

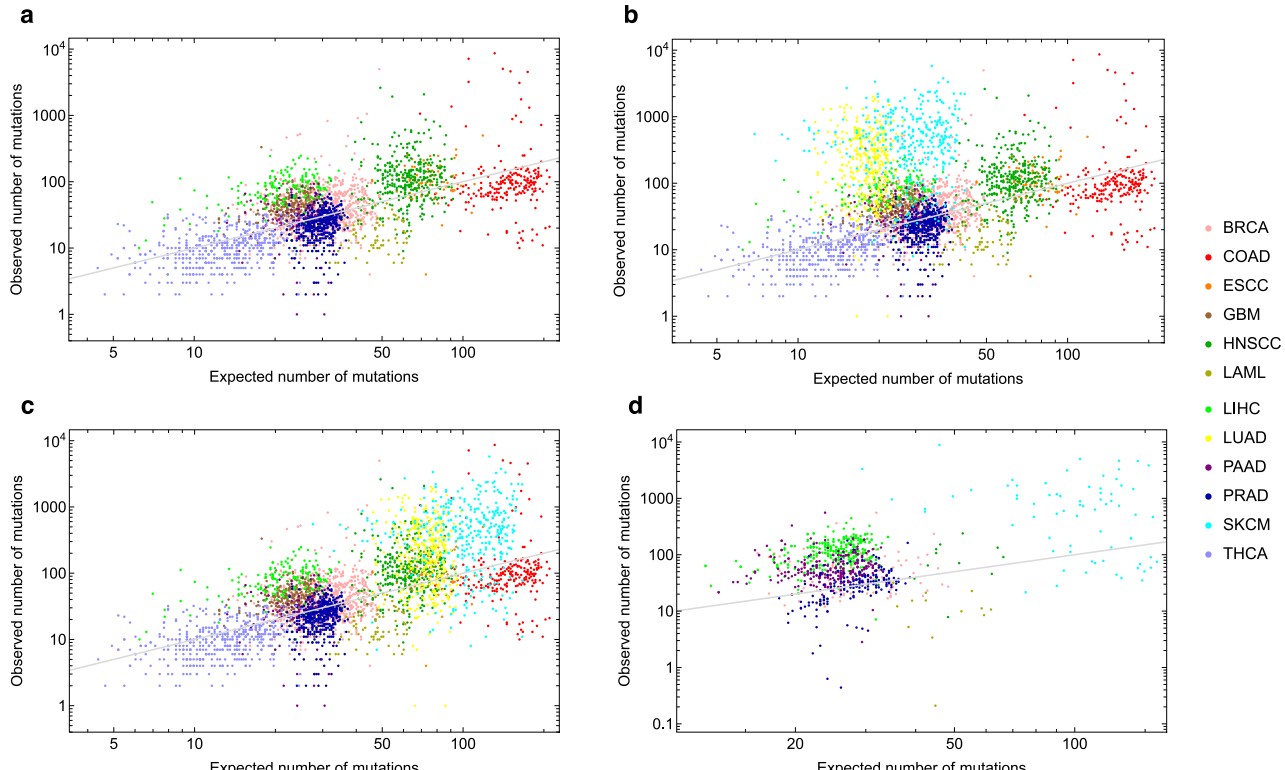

**Fig. 1 | Predicted versus observed number of mutations.** Observed number of somatic mutations found in: **a**–**c** 3608 cancers, for 12 types of cancer (TCGA database), **d** 990 cancers, for 7 types of cancer (ICGC database), versus the corresponding expected number of mutations as predicted by our model (2) for a cancer patient of the same age: **a**–**b** solely due to the endogenous mutational processes, **c**–**d** including the effect of tobacco smoking (LUAD) and UV light (SKCM) as those are the two large outliers observed in Fig. 1b (see Table 2 for the meaning of the abbreviations). The number of drivers for each cancer type is assumed to be $n = 3$ (see Supplementary Fig. S2 for $n = 2$). The gray identity line corresponds to the theoretical case "observed number of mutations" = "expected number of mutations". Source data are provided as a Source Data file.

development phase and yearly number of divisions in healthy and in cancer tissue in Tables 1 and 2 are obtained from other studies taken from the literature. Even the cancer time distribution used in order to estimate the expected cancer occurrence age in formula (2) is only fitted to epidemiological data, specifically the cancer incidence curve.

Finally, to test the robustness of our predictions, we decided to analyze data from the International Cancer Genome Consortium (ICGC), which is another major source of cancer sequencing data. The results—which account also for UV light and are depicted in Fig. 1d—provide again supportive evidence for our model and its predictions.

Figure 2 and Supplementary Fig. S3 provide instead a comparison of the age-dependent number of mutations predicted by our model (orange and red lines), versus the actual number of mutations observed in each patient using sequencing data. Given our time-dependent predictions, we can use them to assess the contribution of the three fundamental etiological risk factors $E$, $H$, and $R$[20] in the following two ways.

First, we can measure how well our predictions fit the actual data in a given cancer type: the more our predictions undershoot the observations, the larger the evidence for important roles played by $E$ and $H$. For example, it is visually clear from Fig. 2c that $E$ (smoking) greatly increases the mutational load in the lungs. In comparison, our predictions for breast and colon (Fig. 2a, b) provide a relatively good fit. Naturally, a good fit does not imply the complete absence of important environmental exposures or inherited factors, but the better the fit, the smaller the evidence of a role for them. Both overshooting and undershooting, if present at the population level, can be caused for example by errors in the estimates we use for the division rate, the mutation probability, or by a difference in the ability of the immune system to fight tumor cells across different organs.

Additionally, when our predictions undershoot the number of mutations actually observed, whether for the whole population or for a subset of patients (points above the gray identity line in Fig. 1 or above the red line in Fig. 2), this can be due to environmental exposures (e.g., smoking or sun exposure), germline mutations (e.g., BRCA1-2 in breast cancer) or genomic instabilities (e.g., microsatellite instability in colorectal cancer) affecting these patients.

We introduce a measure of how well our predictions fit the observations, providing a prediction for how large the effect of $R$ versus $E$ and $H$ factors is. This measure $\delta$ is evaluated as the relative difference between the medians of the set of observed versus expected number of mutations weighted by the age $a_i$ of patient $i$:

$$\delta = \frac{|\,\text{Median}\left(\{\eta^{exp}(a_i)/a_i\}_i\right) - \text{Median}\left(\{\eta_i^{obs}(a_i)/a_i\}_i\right)\,|}{\text{Median}\left(\{\eta_i^{obs}(a_i)/a_i\}_i\right)}. \quad (3)$$

We apply this measure to the case where the number of drivers is one to four (Supplementary Table S2), which is estimated to be the range[21,22]. In general, our predictions provide either a reasonable fit or result in undershooting with the one exception of leukemia (Fig. 2d), where it is possible that our estimate for the rate of cell division is too high or that there is a significant cell division deceleration known to occur with age[9]. We use this same distance $\delta$ to estimate the background mutation rate which minimizes the mean deviation across cancer types (including or not the outliers LUAD and SKCM), as depicted in Fig. 2e. This estimate is equal to 0.024, which is striking as this is very close to the parameter value we originally chose for our model as provided by the available literature ($\mu = 0.03$).

**Table 1 | Parameter estimates and number of somatic mutations (predicted, observed, and in healthy individuals)**

| Cancer name | $D_O$ | $b$ | $b^{(n)}$ | $\rho(\%)$ | $a_m$ | $\eta^h$ | $\eta^c_{n=2}$ | $\eta^c_{n=3}$ | $\eta^o$ |
|---|---|---|---|---|---|---|---|---|---|
| BRCA | 33 | 4.325 | 73 | 10.7 | 56.5 | 8 | 20 | 31 | 23–58 |
| COAD | 27 | 73 | 93.6 | 2.81 | 66 | 145 | 149 | 153 | 69–128 |
| ESCC | 22 | 33.19 | 73 | 0.122 | 57 | 57 | 69 | 74 | 71–143 |
| GBM | 250 | 0.0175 | 73 | 0.246 | 62 | 8 | 9 | 23 | 35–57 |
| HNSCC | 24 | 21.5 | 73 | 0.935 | 60 | 39 | 54 | 62 | 69–148 |
| LAML | 27 | 12 | 73 | 0.330 | 61 | 23 | 39 | 48 | 11–22 |
| LIHC | 31 | 1.1 | 73 | 0.802 | 57 | 3 | 10 | 22 | 55–119 |
| LUAD | 30 | 0.07 | 73 | 0.257 | 67 | 1 | 3 | 19 | 81–354 |
| PAAD | 31 | 1 | 73 | 1.08 | 65 | 3 | 11 | 24 | 21– 33 |
| PRAD | 27 | 3 | 73 | 10.5 | 62 | 6 | 18 | 30 | 17–32 |
| SKCM | 31 | 2.4875 | 73 | 1.67 | 58 | 5 | 15 | 27 | 182–679 |
| THCA | 25 | 0.0875 | 73 | 1.17 | 46 | 1 | 3 | 13 | 7–13 |

For 12 cancer types and associated tissue (see Table 2 for the meaning of the abbreviations), (a) *parameter estimates obtained from the literature* (columns 1–4): $D_O$ number of divisions during the tissue development phase, $b$ and $b^{(n)}$ yearly number of divisions in a healthy and in cancer tissue, $\rho$ proportion of the population that gets cancer by age 80, (b) *data* (columns 5 and 9): $a_m$ median age of the TCGA database cancer patients, $\eta^o$ interquartile interval of the actual observed number of somatic mutations found in the tissue of cancer patients aged $a_m - 5$ to $a_m + 5$, (c) *values computed from our model* (columns 6–8): $\eta^h = \eta^h(a_m)$ (resp. $\eta^c = \eta^{exp}(a_m)$) expected number of somatic mutations found in the tissue of a healthy individual (resp. cancer patient) aged $a_m$ in the absence of $E$ and $H$ factors (formulas (1)-(2)). The expected number of mutations $\eta^c$ in a cancer patient is given both for $n = 2$ and $n = 3$.

Second, within each cancer type, we can stratify patients based on how far their mutational load is from our predictions. For example, in colon (Fig. 2b), there are several patients that are clear outliers with respect to the others, due to the presence of microsatellite instability in their cancer[21]. In fact, our results can be used for risk stratification of individuals that have no cancer yet: this is done by considering the distance of the total number of mutations present in an individual's organ from the expected value of mutations of a healthy individual of the same age as provided in formula (1). Naturally, this constitutes the first attempt to stratify individuals and more work is required to both assess and improve its potential.

In all this analysis, we assumed that the number $n$ of driver mutations required to develop cancer is set to $n = 3$ across all cancer types. Rather than picking a specific $n$ value for each cancer type, we increase the robustness of our analysis by considering the effect of a varying $n$ ($n$ being typically equal to two or three in most cancers[21]), going from a fixed $n = 2$ in Supplementary Fig. S2 to a fixed $n = 3$ in Fig. 1.

## Weibull distribution for the time to cancer

Thanks to our mathematical model, we can evaluate the age at which cancer arrives in a patient, assuming that $n$ driver mutations are required to develop cancer. The cancer appearance time is in this model a random variable whose distribution can be computed under simplifying assumptions detailed in the Methods section. We prove in particular that, in the case of a cancer with a negligible risk of any driver mutation appearing before birth, the time to cancer approximately follows a Weibull distribution $\mathcal{W}(n, c_n N u^n b^n)$, where $c_n$ is some constant depending neither on the number of tissue stem cells $N$, nor on the proliferation rate $b$ or on the driver gene mutation probability $u$ (of the order of $10^{-7}$, see Supplementary Methods). The probability of getting cancer by age $t \geqslant 0$ is therefore

$$P(t) = 1 - e^{-c_n N u^n b^n t^n}. \tag{4}$$

The cumulative risk (4) as a function of age is given in Fig. 3a for three types of cancer (breast, colon, and lung), while the corresponding probability density function is plotted in Fig. 3b. Note that the latter is depicted over a longer time span than the human life expectancy.

Not surprisingly, the value of the shape parameter of the Weibull is $n$, which is usually >1, yielding the desirable result that its failure rate increases with time, consistent with the fact that cancer incidence behaves like a power-law with respect to age[23].

## A re-analysis of the role of endogenous mutations in cancer risk

In ref. 6 it was shown that the endogenous mutational processes associated with cell divisions could explain a large fraction (2/3) of the variation in cancer risk across cancer types. The high correlation found in the U.S. (Pearson's linear correlation 0.80) was later confirmed in a much larger analysis across the world[7]. However, the slope of that linear correlation was 0.47, i.e., much lower than 1. Clearly, the assessment of the slope using relatively few noisy data points is much more challenging than the simple test for the presence of a correlation. Theoretically, however, we would have expected a slope equal to, or higher than, 1 and not smaller. To understand the reason, recall that this analysis was inspired by a simple mathematical model, where the lifetime risk of cancer in tissue should be related to two variables: (1) $N$, the number of stem cells in that tissue, and (2) $D$, the lifetime number of divisions in the lineage of each of those cells. Now, a doubling in $N$ should approximately double the risk of cancer, thus yielding a slope = 1. And a doubling of $D$, which is comparable to a doubling of time, as cell divisions act like a molecular clock, should yield a slope > 1 in a log-log scale, given the known power-law relationship that cancer incidence has with respect to age. In this work, we aimed to better understand the reasons for that lower-than-expected slope found in ref. 6. A critical observation for the present analysis is that in ref. 6 the two key variables, $N$ and $D$, were considered together as one variable, approximately given by their product $ND$. This is not ideal as their effects are expected to be linear and superlinear, respectively. A better approach is rather to consider their effects on the lifetime risk of cancer separately, in a three-dimensional analysis. Importantly, the analysis in ref. 6 did not account for the fact that different cancer types are estimated to have different numbers of required mutational events, $n$, to get to cancer. The effect of varying $n$ for different cancer types is critical if we want to properly assess the slope of the relationship. In the case of cancers with negligible risk of any driver mutation appearing before birth, $P$ is given by (4) which, given that $P$ is close to 0, can be approximated by $P = c_n N u^n D^n$. We obtain in particular the proportionality relationship

$$P \propto N D^n. \tag{5}$$

Using (5), and cancer-specific values for $n$ from refs. 21, 22 as described in the Supplementary Methods, we obtain Fig. 4. The

**Table 2 | Abbreviations of cancer types (if applicable, following the TCGA study abbreviations)**

| Abbreviation | Cancer name |
|---|---|
| AOS | Arms osteosarcoma |
| BAS | Basal cell carcinoma |
| BRCA | Breast invasive carcinoma |
| CLL | Chronic lymphocytic leukemia |
| COAD | Colorectal adenocarcinoma |
| DUO | Duodenum adenocarcinoma |
| ESCC | Esophageal squamous cell carcinoma |
| GALL | Gallbladder adenocarcinoma |
| GBM | Glioblastoma multiforme |
| HNSCC | Head and neck squamous cell carcinoma |
| HOS | Head osteosarcoma |
| LAML | Acute myeloid leukemia |
| LIHC | Liver hepatocellular carcinoma |
| LOS | Legs osteosarcoma |
| LUAD | Lung adenocarcinoma |
| MEDB | Medulloblastoma |
| OVG | Ovarian germ cell |
| PAAD | Pancreatic ductal adenocarcinoma |
| PAE | Pancreatic endocrine carcinoma |
| POS | Pelvis osteosarcoma |
| PRAD | Prostate adenocarcinoma |
| SKCM | Skin cutaneous melanoma |
| SMAD | Small intestine adenocarcinoma |
| TES | Testicular germ cell |
| THCA | Thyroid carcinoma |
| THF | Thyroid follicular and papillary |
| THM | Thyroid medullary |

improvement in the fitting of the data is evident, with an adjusted R-squared increasing from 0.64 in the original analysis[6] ($P = 4.8 \times 10^{-7}$) to 0.8 ($P = 4.4 \times 10^{-9}$) (see Supplementary Methods for more details and sensitivity analysis). This implies that $R$ factors can now explain 4/5 of the observed variation in cancer risk, i.e. more than the 2/3 reported in ref. [6]. Importantly, the slope of the log-log plot of cancer risk $P$ versus number of stem cells $N$ is now equal to 0.91, close to the desired value of 1. Also, $n$ is included as a power in one of the two terms of the fitted linear model, and plotting $\ln P$ versus $\ln D^n$ yields a slope of 0.29, so that plotting $\ln P$ versus $\ln D$ yields a slope equal to 0.87, close to 1, when taking the typical value of $n = 3$ for the number of required drivers[21]. Thus the observed slopes are much closer than the one observed in ref. [6] to what we would expect from the model. It is not realistic to think that these theoretical values could be obtained because of the presence of noise in the data as well as other missing factors (e.g., immune system effects) still currently missing in the model.

## Discussion

Based on a mathematical model of tumor evolution, a re-analysis of Tomasetti et al.[6] provided an improved fitting of the variation in cancer incidence across cancer types. The number of cell divisions, and the normal endogenous mutational processes ($R$) associated to it, explained about 4/5 of the variation in cancer risk across tissues. When combined with the comparison between the expected number of somatic mutations due to $R$ versus the observed one, these results overall provide further evidence for the major role $R$ plays in cancer etiology. The distance between the lifetime cancer risk of a given tissue

(a point in Fig. [4]) and the expected risk for that tissue (the plane in Fig. [4]) provides a framework for inferring the risk of each cancer subtype that may be contributed by either an $E$ or $H$ factor. Specifically, the larger the distance, the greater the evidence for an $E$ or $H$ factor. And the analysis of the expected versus observed number of mutations provides a further approach for assessing the presence of $E$ and $H$ factors in the etiology of a given cancer type, naturally depending on the quality of the estimates for the key biological parameters used in the model.

We have obtained analytical formulas for a mathematical model of tumor evolution that includes all phases of tissue, from tissue development to cancer, and where the effects of a carrying capacity, different types of cell division, and different types of driver mutations are accounted for. One of the key results of this model is that under some simplifying assumptions the time to cancer is given by a Weibull distribution, providing a simple probability distribution for the timing of a highly complex evolutionary process. It is interesting to note that the Weibull distribution is one of the most typical distributions used in survival analysis and has been used in several previous analyses as a statistical model for cancer incidence[24-28]. However, those studies could not provide more than a basic justification for the use of a Weibull distribution, simply based on the fact that it is commonly used to model failure times. We believe that our findings provide now a more solid foundation, by representing an approximation of a mechanistic mathematical model of tumor evolution that includes the detailed evolutionary dynamics of cell populations in tissue, thus better justifying its use in assessing the timing of cancer.

Our approach has several limitations. One is given by the simplifying assumptions made about the mechanisms behind tumor evolution. For example, the effects of a tumor's microenvironment were not considered. While these effects are certainly important, their inclusion is challenging, given the lack of analytical closed-form solutions for even much simpler mechanistic models, but we believe our work provides a step in that direction. Another limitation is given by the uncertainty surrounding the relevant parameter values. To deal with this, sensitivity analyses were performed and scenarios with conservative estimates were used. However, progress in the estimation of key parameters for each of the tissues considered, like total number of cells, division frequencies, and number of required driver hits, will be required to increase the value provided by mathematical models of tumor evolution. Depending on the quality of these estimates, the comparison of the expected versus the observed mutational load in cancer patients provides a way to assess the evidence for the presence —and strength—of environmental factors in the etiology of a given cancer.

Our model is based upon the fact that cells accumulate mutations during their lifespan and that this constitutes a major role in tumor evolution[8] but possibly not for aging, according to some studies[29]. It is clear that endogenous processes are operative also in post mitotic tissues such as neurons (e.g., ref. [30]). Indeed, our results provide support for that, showing that for glioblastoma (GBM in Supplementary Fig. S3) the amount of mutations accumulating exclusively at cell division are not sufficient to explain the observed mutational load. While we do not claim that endogenous processes operate exclusively at cell division, we believe our results support the idea of cell division to be the major mutational engine.

Our findings have relevant implications for how to reduce cancer mortality. The general consensus achieved on the role that $R$ plays in cancer etiology[8,12,14-16,20] points to the critical role of early detection in cancer, beside the one played by primary prevention, if we want to significantly decrease cancer mortality. The consequences of these findings are therefore not only relevant for a better understanding of the biology and dynamics of tumor evolution and cancer, but also for

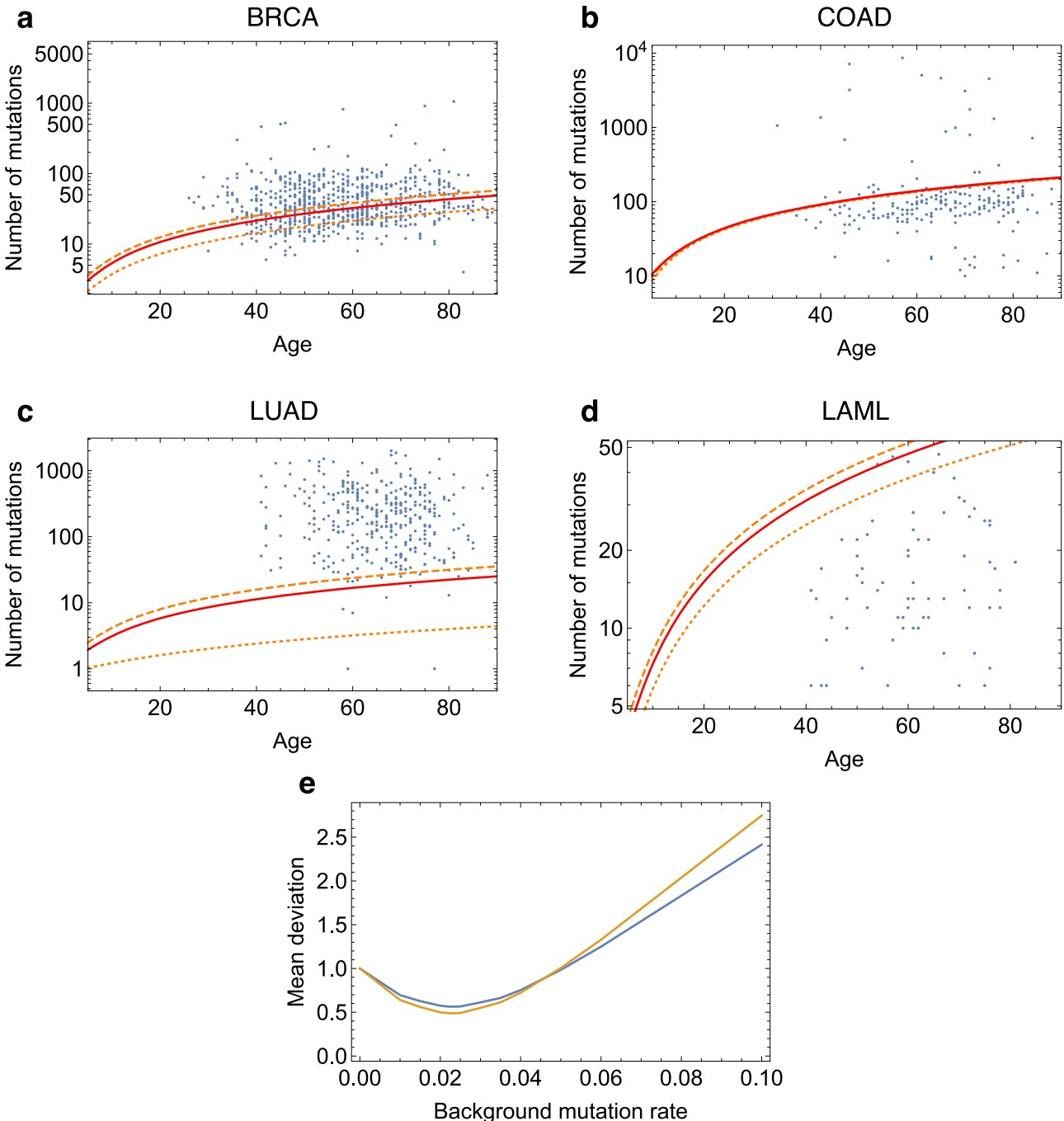

**Fig. 2 | Predicted and observed number of mutations. a–d** Predicted and observed number of mutations as a function of age. Observed number of somatic mutations found in cancers (blue dots) of patients of different ages from the TCGA database, compared with the expected number of mutations solely due to the endogenous mutational processes in patients of the same age, predicted by our model (2), for four types of cancer: breast (BRCA), colon (COAD), lung (LUAD) and leukemia (LAML) (see Supplementary Fig. S3 for other cancer types). The expected number of mutations (2) is given for a number of drivers equal $n = 2$ (dotted orange line), $n = 3$ (dash-dotted orange line) and $n = 4$ (dashed orange line). The solid red line corresponds to the computation for the value of $n$ deduced from ref. 22 and given in Section S6 of the Supplementary Methods. **e** Deviation of the prediction from the observations. Mean value of the deviation $\delta$ given by (3) over all cancer types (blue line) or excluding lung cancer (LUAD) and melanoma (SKCM) (orange line) of the predicted number of mutations from the observed number of mutations, as a function of the background mutation rate. The rate which minimizes the mean deviation is 0.024. Source data are provided as a Source Data file.

indicating what are the most critical approaches in the fight against cancer mortality. After all, if these endogenous processes played only a very minor role in cancer etiology, then almost all of the focus should be on primary prevention and therapy. If, however, they played a major role in cancer causation then only the combination of primary and secondary (i.e., early detection) prevention with therapy will be able to yield large long-lasting reductions in mortality, especially in the presence of an aging population[20].

## Methods

### Review of some key previous models

We review in the Supplementary Methods some key previous mathematical models of tumor evolution[2–4,31–35], from simple two-stage models to more complex models including clonal expansions and different modes of stem cells division. We show that in each model, cancer incidence is, at least approximately: (i) a linear function of $N$ the number of cells at risk, (ii) an exponential or power-law function of $t$

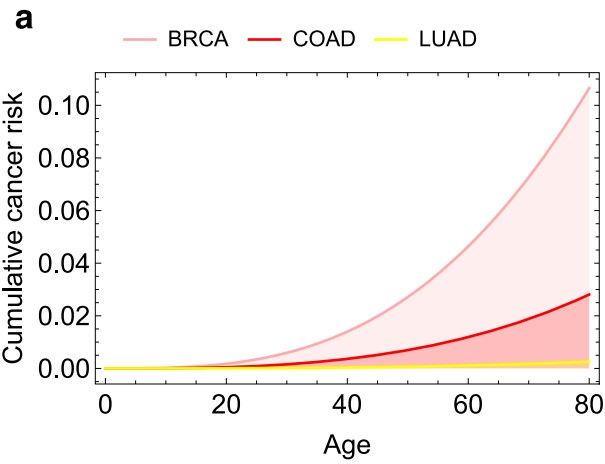

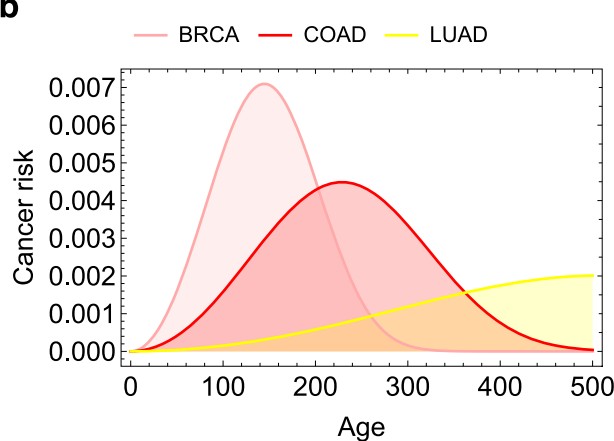

**Fig. 3 | Cancer risk as a function of age for three types of cancer (breast, colon, and lung), as given by the Weibull distribution (4). a** Cumulative distribution function (corresponding to the cumulative cancer risk), **b** probability density function of the Weibull distribution. The parameters of the distribution are such that $n = 3$ and that $P(80) = \rho$, the proportion of the population that gets cancer by age 80 given in Table 1.

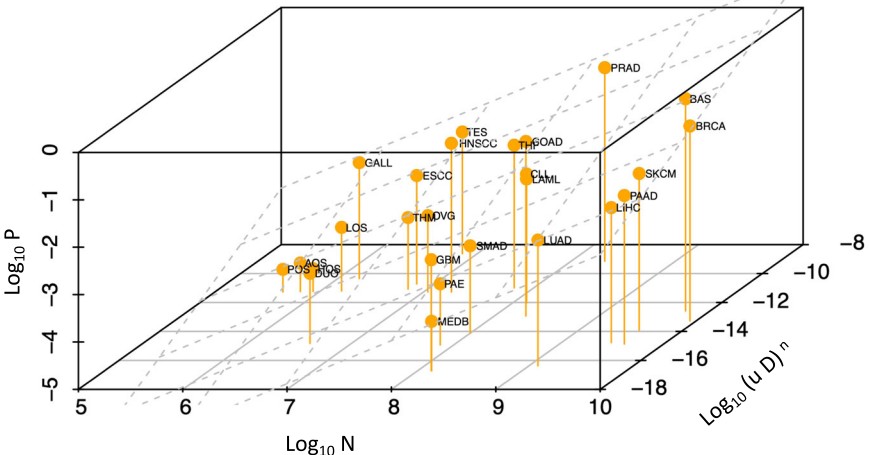

**Fig. 4 | Relationship between lifetime cancer risk $P$, number of stem cells $N$ in a given tissue, and $(uD)^n$ for 26 cancer types, where $D$ is the number of divisions in a stem cell lineage of that tissue, $u$ the driver mutation rate, and $n$ the** **number of required driver mutations.** The plane corresponds to the 3D linear regression surface. See Table 2 for the meaning of the abbreviations. Source data are provided as a Source Data file.

the age of the individual, (iii) a power-law function of $u$ the driver mutation rate, with power of $n$ being the number of stages in the process of tumor formation in the model. These findings are consistent with what is obtained in our mathematical model, for which cancer incidence at time $t$ is approximately $nc_nNu^nb^nt^{n-1}$ (see (4)).

### Analytical mechanistic model of tumor evolution

As in the simulation model presented in ref. 10, the key features of our model are the following. First, we model all temporally distinct phases in the lifetime of a tissue, starting with its development, followed by homeostasis and tumor evolution all the way to, possibly, cancer. Second, we steer clear of the typical yet somewhat unrealistic assumption of exponential tumor growth. Instead, we take into account the early random fluctuations of any newly arising stem cell population, and assume logistic growth following this initial stochastic phase. We recall that logistic growth is made by an initial exponential phase followed by a growth rate that becomes smaller the closer the size of the cell population is to its carrying capacity, to reflect natural constraints like energy resources or spatial limitations (Fig. 5b). Because the clonal expansion may have different names in different tissues (e.g., a nodule in the pancreas or the breast, a polyp in the intestine, etc.), we use the most general term "clone" and refer to it

throughout this paper. Third, it is a mechanistic model, accounting for the birth and death of each stem cell and its different types of cell division: symmetric self-renewal, symmetric differentiation, and asymmetric division (Fig. 5a). We focus exclusively on stem cells as their stem cell progeny is able to carry a dangerous mutation all the way to cancer. To account for the fact that there are generally more frequent self-renewals during the development phase of a tissue, we let the asymmetric division probability $p$ increase through time and reach its maximal value in adulthood once the tissue has been fully developed. Fourth, the model follows the number and types of driver mutations that a stem cell lineage accumulates randomly, distinguishing among three main types. Specifically, each driver mutation confers a selective advantage, the nature of which depends on the cellular pathways affected by the mutation. These pathways can be classified into three core cellular processes: cell survival ($S$), cell fate ($F$), and genome maintenance ($M$) (as described in ref. 10). If the mutation affects the pathway which regulates the cell growth and survival process, termed a $S$ mutation, it translates into an increase of the proliferation rate $b$ in the clone carrying the $S$ mutation. As a result, the growth rate and the carrying capacity of its subsequent logistic growth increase too, as well as its mutation rate per cell and per time unit. If the pathway which regulates the cell fate ($F$) is affected, the

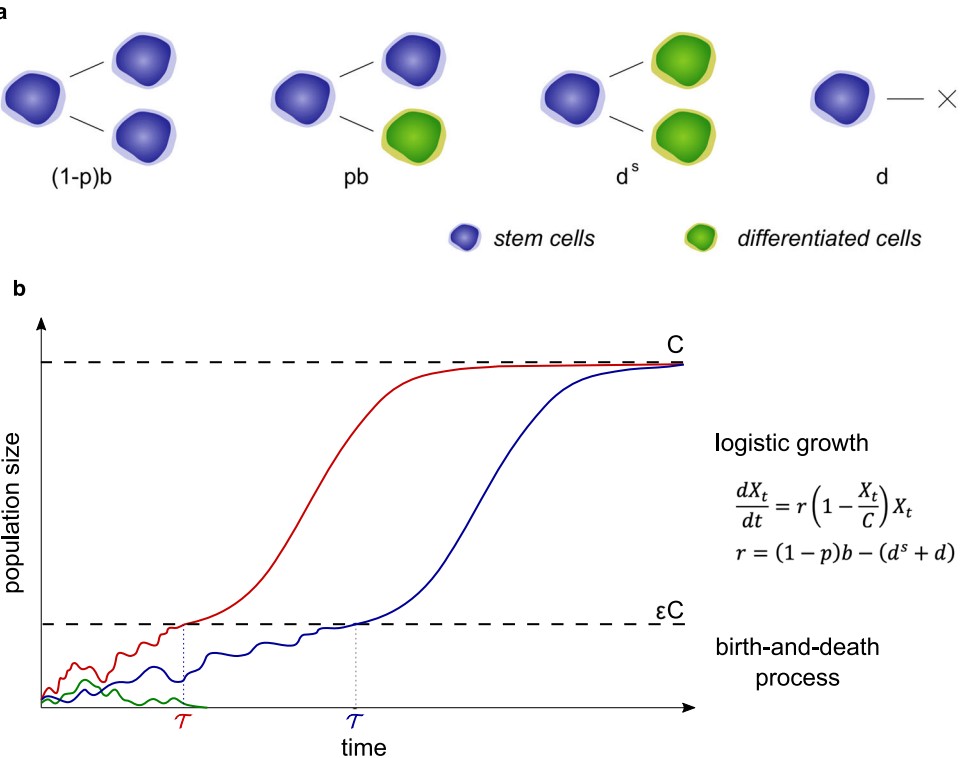

**Fig. 5 | Growth of a tissue stem cell population. a** Rates of self-renewal, asymmetric division, symmetric differentiation, and death of a stem cell. **b** An illustration of the size evolution of three stem cell population trajectories, according to the model described in the Methods section.

balance between cell symmetric self-renewal and asymmetric division is shifted, favoring symmetric self-renewal. This translates in our model into a decrease of the asymmetric division probability $p$, which in turn induces an increase in the mutation rate, growth rate, and carrying capacity of the clone. Finally, if the genomic maintenance ($M$) pathway, which is responsible for DNA repair, is affected, it translates into an increase of the mutation probability $u$ in the clone carrying the $M$ mutation, hence its mutation rate per cell and per time unit. The details of these different effects on the dynamics and mutation parameters of a clone can be found in the Supplementary Methods. We are then able to compute analytically the lifetime risk of getting cancer in a given tissue for an individual.

**Analytical expression of cancer risk**

To provide intuition for the key ingredients of the model we describe the typical process, as depicted in Fig. 5. We model the growth of the tissue stem cell population with a process $N_t$ composed of three phases ($\psi_0$: from conception to birth, $\psi_1$: youth, $\psi_2$: adulthood), growing as $t$ increases from size 0 to $N$, where $N$ is the maximum stem cell population size reached when the tissue is in homeostatic equilibrium. To account for the fact that the rates of the various cell division types differ greatly depending on the specific phase the tissue is in, we assume that during phase $\psi_i$ the proliferation rate and asymmetric division probability are equal to $b_i$, $p_i$. Next, for each newly arising clone, we assume the following pattern. The clonal population starts small hence the cells do not compete with each other, and we can assume that they initially evolve independently of each other. Consequently, their number follows at first a random birth-and-death process, until it potentially reaches a fraction $\varepsilon$ of the clone carrying capacity $C$, where $C$ corresponds to the maximum stem cell population size that the clone can sustain, and $\varepsilon > 0$ is relatively small, but sufficient to ensure survival. Specifically, each stem cell renews itself at time-dependent rate $(1 - p)b$, asymmetrically differentiates at rate $pb$, symmetrically differentiates at rate $d^s$ or dies at rate $d$ (Fig. 5a). The clonal stem cell population thus initially evolves as a birth-and-death

process starting with one cell, with birth rate $(1 - p)b$ and death rate $d^s + d$, hence with growth rate $r = (1 - p)b - (d^s + d)$. In the case the birth-and-death process does not become extinct and the clone reaches size $\varepsilon C$, the effects of the law of large numbers render the growth process deterministic and we assume that from this time onwards the clonal population grows logistically with carrying capacity $C$ and growth rate $r$, as illustrated in Fig. 5b (see details in the Supplementary Methods).

For a given tissue and a given cancer type, we define $\mathcal{C}$ as the set of all possible sequences of $n$ driver mutations $V_1 \cdots V_n$ leading to this cancer, where each mutation $V_i$ can be of type $S$, $F$ or $M$ as defined in the description of our analytical mechanistic model. For instance, $n = 3$ and $\mathcal{C}$ is the set of triples with at least one $S$ and one $F$ mutation. An individual will then have developed cancer by age $t$ if at least one of its cell lineages carries such a $\mathcal{C}$-sequence of mutations (Supplementary Fig. S1). Assuming that any newly arising clonal population evolves independently of the existing cell populations, the probability $P(t)$ of getting cancer by age $t$ can be computed by means of the associated point process for which an event corresponds to the appearance of a clone of "survival" size leading to cancer before $t$. Survival size means here large enough to ensure survival of the population to stochastic extinction. Let $-a_0$ be the time of conception and 0 the time of birth. For any $t \geq -a_0$, the cancer risk $P(t)$ which corresponds to the probability that at least one event occurs before $t$ is obtained as

$$
\begin{aligned}
P(t) \quad &= 1 - \mathbb{P}(\text{no event before } t) \\
&= 1 - \exp\left(-\int_{-a_0}^{t} \lambda(s)\,ds\right),
\end{aligned}
\tag{6}
$$

where $\lambda(s)$ is the intensity of the point process, namely the infinitesimal rate at which events are expected to occur around $s$. Let $P_{V_1}(t - s)$ the probability for a clone carrying a first mutation $V_1$ of type $S$, $F$ or $M$, to lead to cancer before time $t$, given that the $V_1$-clone reached a survival size at $s$, and $\mu_{V_1}(s)$ the appearance rate of such a clone. The intensity

$\lambda(s)$ is then given by

$$\lambda(s) = \sum_{V_1 \in \{S,F,M\}} \mu_{V_1}(s) P_{V_1}(t - s). \tag{7}$$

A $V_1$-clone can reach survival size at time $s$ only if mutation $V_1$ appeared at some time $-a_0 \leqslant z \leqslant s$. We assume that the driver gene mutations appear randomly in the stem cell populations, at rate proportional to the population size (given by $N_z$ for the tissue cell population, and by the logistic growth for a clone). Denoting by $u$ the driver gene mutation probability per cell division, we deduce from what precedes that the driver mutation rate per cell lineage and per time unit is $(2 - p)ub$. Note that the values of $p$ and $b$ depend on the specific phase the tissue is in, and we, therefore, denote by $(2 - p(z))ub(z)$ the driver mutation rate at time $z$. Consequently, the appearance rate of a surviving $V_1$-clone in the tissue cell population is

$$\mu_{V_1}(s) = \int_{-a_0}^{s} (2 - p(z))ub(z)N_z \pi_{V_1} \rho_{V_1} f_{V_1}(s - z)dz, \tag{8}$$

where $\pi_{V_1}$ is the probability that a driver mutation is of type $V_1$, $\rho_{V_1}$ the probability for a $V_1$-clone of reaching a survival size, and $f_{V_1}$ the latency period distribution to reach this size. We then compute in the same manner $P_{V_1}$ which involves probabilities of the form $P_{V_1 V_2}$ for clones carrying an additional mutation $V_2$. Iterated $n$ times, the formula finally includes the probabilities $P_{V_1 \cdots V_n}$, which are 1 if $V_1 \cdots V_n \in \mathcal{C}$ and 0 otherwise. We, therefore, obtain an explicit formula for the probability $P(t)$, and by extension of the lifetime cancer risk $P$ if $t$ is chosen as the life expectancy. We refer to the Supplementary Methods for more details on these computations.

## Distribution of the time to cancer

Let $T_n$ be the cancer appearance time, which is a random variable with values in $[-a_0, +\infty)$ and with cumulative distribution function $P(t)$ given by (4). In order to isolate the effect on the cancer time distribution of some key parameters such as the driver mutation probability $u$, the cell division rate $b$ and the total number of stem cells $N$, we neglect both the initial (random) and growth (deterministic) phases of each cell population, assuming that any driver mutation will give rise to a clone of size equal to its carrying capacity. We also suppose that the proliferation rate and asymmetric division probability are constant across youth and adulthood ($b_1 = b_2 = b$, $p_1 = p_2 = p$), and equal to $b_0$, $p_0$ between conception and birth, i.e. between time $-a_0$ and 0. Under these simplifying assumptions, we prove (see Supplementary Methods) that $P(t) = 1 - e^{-\Lambda(t)}$ is of the form

$$\Lambda(t) = \frac{1}{a_0^{n+1}} c_0 N u^n b_0^n (a_0 + t)^{n+1}, \qquad -a_0 \leqslant t \leqslant 0,$$
$$\Lambda(t) = N u^n \left( b_0^n (c_0 + c_1 t + \ldots + c_{n-1} t^{n-1}) + b^n c_n t^n \right), \quad t \geqslant 0. \tag{9}$$

From this we deduce that the random variable $T_n^+$, which is the time to cancer $T_n$ conditional on the event that cancer appears after birth, is distributed as the minimum of $n$ independent Weibull random variables:

$$T_n^+ \sim \min(W_1, \ldots, W_n), \tag{10}$$

where $W_k \sim \mathcal{W}(k, c_k N u^n b_0^n)$ $(k < n)$ and $W_n \sim \mathcal{W}(n, c_n N u^n b^n)$. The notation ~ means "follows the distribution", and we use the convention that a Weibull distribution $\mathcal{W}(\alpha, \beta)$ with shape parameter $\alpha$ and scale parameter $\beta$ has cumulative distribution function $t \mapsto 1 - e^{-\beta t^\alpha}$. Similarly, the time of cancer appearance, conditional on the event that cancer occurs between conception and birth and denoted by $T_n^-$, satisfies $T_n^- \sim W_0$, where $W_0$ follows a translated and truncated Weibull distribution with values in $[-a_0, 0]$. Each $W_k$ can be thought as the time

of cancer appearance when exactly $0 \leqslant k \leqslant n$ driver mutations occur after birth, and therefore $n - k$ before birth.

Finally, we show that in cancer types for which the probability of driver mutations occurring before birth is negligible compared to the cancer risk, $T_n \sim \mathcal{W}(n, c_n N u^n b^n)$. We recall that the Weibull distribution $\mathcal{W}(n, \beta)$ (which under the alternative standard parameterization has scale $\lambda := \beta^{-1/n}$) has the nice property to be the maximum entropy distribution for a non-negative real random variable $X$ with an expected value of $X^n$ equal to $\lambda^n$. The proof of all the preceding results is provided in the Supplementary Methods.

## Predicted number of mutations

The expected number of somatic mutations found in a cell of a healthy individual of age $a$ is given by formula (1). In contrast, if $T_1, \ldots, T_n$ denote the hitting times of the $n$ required driver mutations and $b^{(i)}$ the cell division rate after $i$ driver hits, then the expected number of somatic mutations in a cancer cell of a patient of age $a$ is

$$\tilde{\eta} = \mu \left( D_0 + bT_1 + b^{(1)}(T_2 - T_1) + \ldots + b^{(n)}(a - T_n) \right). \tag{11}$$

While the above formula can be applied to single-cell sequencing data, in the case of bulk sequencing we only observed the mutations accumulated in the winning lineage of the first cancer cell, born at time $T_n$, as all the subsequent mutations will not be called by the sequencer. In that case, the appropriate formula is

$$\eta = \mu \left( D_0 + bT_1 + \ldots + b^{(n-1)}(T_n - T_{n-1}) \right). \tag{12}$$

For a given cancer and a cancer patient of age $a$, we compare the number of somatic mutations $\eta^{obs}(a)$ observed via bulk sequencing of a cancer sample of that individual, with the expected value of $\eta$ conditioned on the event that cancer appeared during the time interval $[a - c, a]$ (see Supplementary Methods for the choice of $c$) $\eta^{exp}(a) = \mathbb{E}(\eta | a - c \leqslant T_n \leqslant a)$. Assuming for simplicity that the driver hitting times satisfy $T_i = \frac{i}{n} T_n$, it follows from the expression of $\eta$ that $\eta^{exp}(a)$ is given by (2), where

$$a_n^* = \mathbb{E}(T_n | a - c \leqslant T_n \leqslant a). \tag{13}$$

For cancers with a negligible risk of any driver mutation appearing before birth, the probability distribution of the cancer occurrence time $T_n$ is given by (4). In this case

$$a_n^* = \beta^{-\frac{1}{n}} \frac{\gamma\left(1 + \frac{1}{n}; \beta(a - c)^n, \beta a^n\right)}{\gamma(1; \beta(a - c)^n, \beta a^n)}, \tag{14}$$

with $\beta := c_n N u^n b^n$ and $\gamma(z; a, b) := \int_a^b t^{z-1} e^{-t} dt$. Parameter $\beta$ can be deduced from epidemiological data, for instance from the proportion of the population that gets cancer by age 80 (Table 1): $\beta = -\ln(1 - \rho)/80^n$. Moreover, if $s$ is the fitness advantage of a $S$ mutation and if $k$ out of the first $i$ driver hits are of type $S$, then $b^{(i)} = (1 + s)^k b$. From the final value $b^{(n)}$, which corresponds to the division rate in tumors and which can be estimated (Table 1), we deduce $s$ by averaging over the different scenarios of occurrences of $S$ mutations among the $n$ drivers (see Supplementary Methods for more details), which finally leads to an explicit expression of $\eta^{exp}(a)$.

## Observed number of mutations

We downloaded somatic exomic mutational data from the TCGA Bioportal (https://portal.gdc.cancer.gov) and whole-genome sequencing data from the ICGC database (https://dcc.icgc.org/releases/PCAWG). Only ICGC datasets that were not present in the TCGA database were used. The total number of mutations across the exome was inferred by adding all the point mutations (divided by 100 in the case of whole-genome sequencing). This number then provided the

input for the analysis presented in Figs. 1–2, Table 1, and Supplementary Fig. S2–S3, for patients for which the age was specified. Source data including the patient ID are provided with this paper.

## Reporting summary

Further information on research design is available in the Nature Portfolio Reporting Summary linked to this article.

## Data availability

The results published here are based on data generated by The Cancer Genome Atlas Research Network (https://www.cancer.gov/tcga) and the International Cancer Genome Consortium (https://dcc.icgc.org). There are no restrictions on access to these data, which are all publicly available. The data generated for this study ("expected number of mutations" in Figs. 1–2, Table 1, and Supplementary Figs. S2–S3) are obtained by the direct application of formula (2). Source data are provided with this paper.

## Code availability

A code implementing formula (2) in the Wolfram Language is provided as Supplementary Software. The analysis leading to Fig. 4 has been performed with the software R (version 4.1.2) and is fully described in the Supplementary Methods.

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

## Acknowledgements

We thank Isaac Kohane for useful comments and suggestions. We thank the John Templeton Foundation (#61471, C.T.) and The Robert & Lynda Carter Altman Family Foundation Research Fund (S.P. & C.T.) for providing funding for this research.

## Author contributions

C.T. conception; S.P., A.L., and C.T. research design and mathematical modeling; S.P. and C.T. data analysis; S.P., A.L., and C.T. writing of the manuscript.

## Competing interests

The authors declare the following competing interests: under a license agreement between Exact Sciences Corp. and Johns Hopkins University,

C.T. is entitled to royalty distributions. C.T. is a member of the Scientific Advisory Board of PrognomiQ, Inc. C.T. is also a paid consultant to Bayer AG. These arrangements have been reviewed and approved by Johns Hopkins University in accordance with its conflict of interest policies. S.P. and A.L. declare no competing interests.
