## [Peer Review File · Nature Communications]

REVIEWER COMMENTS

Reviewer #1 (Remarks to the Author): Expert in mathematical modelling and cancer evolution

The authors present a mechanistic, evolutionary mathematical model which captures the key dynamics in mutational processes through development, homeostasis and cancer initiation. The goal of the manuscript is to go beyond previous similar models by providing an analytical justification for the Weibull distribution of cancer incidence, and to quantify the amount of variation in cancer risk which can be attributed to random processes. The re-analysis of the role of endogenous mutations in cancer risk states that lifetime risk of cancer in a tissue should be related to two variables: 1) N , the number of stem cells in that tissue and 2) D , the lifetime number of divisions in the lineage of each of those cells. Authors correct previous literature by noting that the role of N should be linear, and the role of D should be super-linear (log-log).

The central result appears to be that R factors ("bad luck") can explain 4/5ths of the observed variation in cancer risk, which is an increased value when compared to previous estimated (i.e. reference 6). However, this central result is surprising, given the fact that the authors base this finding on improved fits to data, but the data shown in Figure 1 and 2 exhibit quite extensive deviation from model-fits, and the authors show two possible models in Figure 2.

While this is useful and informative, it appears to be a moderate extension upon previous mathematical formulations with no new data and little attention given to the predictions provided using the model. This is interesting and careful work, but would be more appropriate for a more mathematical audience.

Major comments:

Q1: can this framework be used to infer the risk of each cancer subtype which can be contributed to either E or H factors?

The authors claim that the majority of cancer risk can be attributed to endogeneous mutational processes. However, the impact of this statement is unknown, as it simply provides an updated figure from a previous publication from the same authors. The paper's impact would be improved by providing some predictive value, beyond the scope of previous work. Note: this is mentioned in the Discussion, but no analysis is performed.

Q2: why are the slopes for 1) P vs stem cells N and 2) P vs log(D) not equivalent to 1?

Although the authors' framework admirably approaches a value of unity in these two factors (discussed in line 188-190 on page 6), some discussion should be devoted to why values continue to diverge from one.

Q3: the modeling analysis provides the expected value of cancer incidence, but little discussion is devoted to variation in incidence implied by the model (which should be shown in Figure 1 and 2).

The Weibull distribution is a time-dependent function, and some attempt to visualize variation in cancer risk should be added to figures 1 and 2.

Q4: why is the number of drivers not inferred by best fit in Figure 1, and compared to known biology?

The mathematical model makes some interesting predictions (e.g. the number of drivers inferred per disease type), and the authors should make definitive statements about these inferences in order to use the model to generate or test hypotheses.

Minor comments:

1) Page 10, line 313: what is an "S and F mutation"? – please define.

2) Page 10, line 324: what is an "M mutation"? – please define.

Reviewer #3 (Remarks to the Author): Expert in genomics and somatic evolution in healthy tissues and cancer

In this article, Penisson and colleagues report on a mathematical model to predict tumour evolution and cancer risk. They suggest that this model may lead to estimating cancer risk and early detection.

The scope of the work is of high interest and unique towards understanding the distinct factors that affect clonal evolution and cancer risk that are still uncertain. I commend the team for their efforts in

creating such model, which would be useful for the entire cancer research community. Regarding the observations derived from this resource, some of the findings reported here are consistent with prior reports that are mainly contributed by the authors' previous works.

However, the study would strongly benefit from significant additional clarifications given the approaches here relative to prior work in this domain. In addition, the novel findings in this study would need additional work such as testing on different dataset, as it is not certain that the claims are supported by the data as presented. Below are some of my main concerns.

How exactly priors such as R value and number of drivers was determined is unclear, this is a critical issue given that number of drivers forms a substantial portion of their predictions.

How does the new model lend itself to be a potential tool for cancer risk stratification? The authors are overstating the potential translational relevance of their findings. In Figure 2, the authors showed that their predication changes with the number of drivers (n=2 or n=3). What happen to tumours with no known drivers?

The predication of expected number of mutations does not seem to work exceptionally well. The fact that the model is particularly doing worse in predicting mutations in melanoma and lung cancers, that are mostly affected by the exogenous factors such as UV and smoking, raises concerns that the model is too simplistic and requires a more complex version to reflect the complexity of cancers and clonal evolution in different tissues. Additionally, my worry is that these two cancers have high burden of mutations which might particularly pronounce the shortcoming of the model, and this is less visible in cancers with low mutation burden.

It would be informative to know how this model holds if applied on different data other than TCGA such as PCAWG.

Currently, the model is based on cell replication error. Although this was thought be the main player in cancer evolution, recent studies such Robinson et al., Nature 2021 (<https://www.nature.com/articles/s41588-021-00930-y>) have shown that human tissues can physiologically tolerate ubiquitously elevated mutation burdens. Such burdens do not support a model in which all features of cancer and ageing are attributed to widespread cell malfunction, directly resulting in high somatic mutation burdens.

Moreover, recent work by Abascal et al., Nature 2021 (<https://www.nature.com/articles/s41586-021-03477-4>) has showed that endogenous signatures attributed to ageing and cancer are also operative in

post-mitotic tissues such as neurons. Hence, cell replication error cannot be the main contributor to ageing and cancers. Therefore, this raises concerns whether the current choices for this model, which are heavily weighted towards the cell replications errors in cancer, can predict a realistic cancer risk.

Minor concerns

The language used to describe the model is not particularly accessible to the readers of this journal. Some part can be simplified.

Figure 1 some cancer types are indistinguishable, as they have same or very similar colours.

Authors' Replies to Reviewers' Comments

We would like to thank the reviewers for the useful comments, which have resulted in important improvements of the manuscript, as described in detail in our point-by-point replies below (in blue).

Reviewer #1 (Expert in mathematical modelling and cancer evolution)

The authors present a mechanistic, evolutionary mathematical model which captures the key dynamics in mutational processes through development, homeostasis and cancer initiation. The goal of the manuscript is to go beyond previous similar models by providing an analytical justification for the Weibull distribution of cancer incidence, and to quantify the amount of variation in cancer risk which can be attributed to random processes. The re-analysis of the role of endogenous mutations in cancer risk states that lifetime risk of cancer in a tissue should be related to two variables: 1) N , the number of stem cells in that tissue and 2) D , the lifetime number of divisions in the lineage of each of those cells. Authors correct previous literature by noting that the role of N should be linear, and the role of D should be super-linear (log-log).

The central result appears to be that R factors ("bad luck") can explain 4/5ths of the observed variation in cancer risk, which is an increased value when compared to previous estimated (i.e. reference 6). However, this central result is surprising, given the fact that the authors base this finding on improved fits to data, but the data shown in Figure 1 and 2 exhibit quite extensive deviation from model-fits, and the authors show two possible models in Figure 2.

We thank the reviewer as we have now more carefully addressed in the main text the important point of the extensive deviations observed in Fig 1-2, and we have also produced some new analysis and added Fig 1c. We would like to note that the analysis resulting in the improved fit in cancer risk (Fig 4) is different from the analysis presented in Fig 1, 2, S2, S3, as one is about the probability of getting cancer in a given organ, while the other is about the total number of mutations expected in a given organ at a given age.

While this is useful and informative, it appears to be a moderate extension upon previous mathematical formulations with no new data and little attention given to the predictions provided using the model.

We respectfully believe the new mathematical content to represent substantial progress with respect to the previous formulation of reference 6.

This is interesting and careful work, but would be more appropriate for a more mathematical audience.

We have now moved a substantial amount of the mathematical modeling to the Methods section.

Major comments:

Q1: can this framework be used to infer the risk of each cancer subtype which can be contributed to either E or H factors? The authors claim that the majority of cancer risk can be attributed to endogenous mutational processes. However, the impact of this statement is unknown, as it simply provides an updated figure from a previous publication from the same authors. The

paper's impact would be improved by providing some predictive value, beyond the scope of previous work. Note: this is mentioned in the Discussion, but no analysis is performed.

We thank the reviewer for this point. We have now added in the first paragraph of the discussion the following explanation: "The distance between the lifetime cancer risk of a given tissue (a point in Fig 4) and the expected risk for that tissue (the plane in Fig 4) provides a framework for inferring the risk of each cancer subtype that may be contributed by either an E or H factor. Specifically, the larger the distance, the greater the evidence for an E or H factor."

To further improve the impact of the paper, we have also added in the main text and in the Supplementary Information a new measure of the deviation of the expected number of mutations from the observed number of mutations for each cancer type, providing a prediction for how large the effect of R versus E and H factors (Suppl Table S2) is. While we stated that endogenous mutational processes play a "major role" in cancer risk, this does not necessarily imply the majority of it.

Another key result we have now added is that we used this new distance to estimate the background mutation rate which minimizes the mean distance across cancer types (including or not the outliers LUAD and SKCM, Fig 3c). That estimate is equal to 0.024, which is striking as this is very close to the parameter value we originally chose for our model as provided by the available literature.

Q2: why are the slopes for 1) P vs stem cells N and 2) P vs log(D) not equivalent to 1?

Although the authors' framework admirably approaches a value of unity in these two factors (discussed in line 188-190 on page 6), some discussion should be devoted to why values continue to diverge from one.

As the reviewer mentions, we think this is quite satisfying to see that the slope approaches a value of unity. The slopes are never going to be =1 simply due to the presence of noise in the data as well as other missing factors (immune system effects) still currently missing in the model.

Q3: the modeling analysis provides the expected value of cancer incidence, but little discussion is devoted to variation in incidence implied by the model (which should be shown in Figure 1 and 2). The Weibull distribution is a time-dependent function, and some attempt to visualize variation in cancer risk should be added to figures 1 and 2.

Fig 1 and 2 are not about cancer incidence but rather about the total numbers of mutations expected and observed and so it would not be possible to show the variation in cancer incidence in those figures. At the same time, by including the genomic data of each patient, Fig 1 and 2 are able to show the large variation observed across different patients and different cancer types.

We have followed the reviewer's suggestion by including a graphic comparison of the cancer risk for three cancer types (breast, colon, and lung), as a function of time (Figure 3).

Q4: why is the number of drivers not inferred by best fit in Figure 1, and compared to known biology? The mathematical model makes some interesting predictions (e.g., the number of drivers inferred per disease type), and the authors should make definitive statements about these inferences in order to use the model to generate or test hypotheses.

We thank the reviewer for this suggestion. We have included a new measure, a new analysis, a new figure (Fig 2e) and a new table (Suppl Table S2), where we now provide clear statements and predictions on background mutation rate, on the evidence of E and H factors in a cancer type, on risk stratification among patients, and on the number of drivers for each cancer type. Given that the predictions / estimates we provide depend on several covariates (e.g. in Suppl Table S2, number of drivers, presence or not of E and H factors...), we would prefer to leave to the reader the task of choosing their preferred covariates' values in order to generate *definitive* statements. For example, in determining which tissue is most affected by the environment using Suppl Table S2, the reader will have to choose what n, the number of drivers, should be. However, we are happy to satisfy further requests of the reviewer if deemed necessary.

Minor comments:

- 1) Page10, line 313: what is an “S and F mutation”? – please define.
- 2) Page 10, line 324: what is an “M mutation”? – please define.

We modified the text to recall the reader the meaning of the different mutation types S, F and M.

Reviewer #3 (Expert in genomics and somatic evolution in healthy tissues and cancer)

In this article, Penisson and colleagues report on a mathematical model to predict tumour evolution and cancer risk. They suggest that this model may lead to estimating cancer risk and early detection.

The scope of the work is of high interest and unique towards understanding the distinct factors that affect clonal evolution and cancer risk that are still uncertain. I commend the team for their efforts in creating such model, which would be useful for the entire cancer research community. Regarding the observations derived from this resource, some of the findings reported here are consistent with prior reports that are mainly contributed by the authors' previous works.

We are delighted that Reviewer #3 recognized the relevance of the model and its findings in this study.

However, the study would strongly benefit from significant additional clarifications given the approaches here relative to prior work in this domain. In addition, the novel findings in this study would need additional work such as testing on different dataset, as it is not certain that the claims are supported by the data as presented. Below are some of my main concerns.

How exactly priors such as R value and number of drivers was determined is unclear, this is a critical issue given that number of drivers forms a substantial portion of their predictions.

With respect to the predictions provided in Fig 1-2 and associated supplementary figures, the results were provided for both n=2 and n=3, as those are expected to be the most probable number of required drivers (Tomasetti et al. PNAS 2015). As for the analysis leading to Fig 4 in the main text, the choice of the number of drivers is detailed in the SI and was obtained from Iranzo et al. PNAS 2018, and a sensitivity analysis is also provided there.

In addition, we have now included a new analysis and a table where we provide an inference on the number of drivers based on the fit of the data (see Suppl Table S2 and SI for details).

And with respect to the choice of the background somatic mutation rate μ , the details are provided in the main text (ref 5) and in the SI. Moreover, we now prove that the background mutation rate which minimizes the deviation of the predicted number of mutations from the observations is very close to this value μ (Fig 3c).

How does the new model lend itself to be a potential tool for cancer risk stratification? The authors are overstating the potential translational relevance of their findings.

We thank the reviewer for his important comment that has brought us to develop a more precise way to quantify two effects: 1) the evidence for each cancer type for the presence or not of important environmental or inherited factors contributing to that cancer, 2) the stratification among patients. Specifically, starting in line 160 of the main text we say “First, we can measure how well our predictions fit the actual data in a given cancer type: the more our predictions undershoot the observations, the larger the evidence for important roles played by E and H. For example, it is visually clear from Figure 2c that E (smoking) greatly increases the mutational load in lungs. In comparison, our predictions for breast and colon (Figure 2a and 2b) provide a relatively good fit. Naturally, a good fit does not imply the complete absence of important environmental exposures or inherited factors, but the better the fit, the smaller the evidence of a role for them. We introduce a measure of how well our predictions fit the observations (see Supplementary Methods for methodology), providing a prediction for how large the effect of R versus E and H factors is.”

And then, from line 178 we talk specifically about stratification and we tone down our statement: “Second, within each cancer type, we can stratify patients based on how far their mutational load is from our predictions. For example, in colon (Figure 2b), there are several patients that are clear outliers with respect to the others, due to the presence of microsatellite instability in their cancer. In fact, our results can be used for risk stratification of individuals that have no cancer yet: this is done by considering the distance of the total number of mutations present in an individual’s organ from the expected value of mutations of a healthy patient of the same age as provided in formula (4). Naturally, this constitutes a first attempt to stratify individuals and more work is required to both assess and improve its potential.”

In Figure 2, the authors showed that their predication changes with the number of drivers (n=2 or n=3). What happen to tumours with no known drivers?

We provided our predictions by considering both the case n=2 and n=3. One of the reasons why we did not include certain cancer types in our analysis is because we didn’t have any available estimate for the number of drivers present in those cancer types. One possibility for solid cancers where the number of drivers is not known would be to assume the typical 2-3 drivers (Tomasetti et al. PNAS 2015). No matter whether the number of drivers is known are not, we work under the assumption that it is the same number of drivers for each patient of a given cancer type, as estimated by Iranzo et al. PNAS 2018 (see SI).

The predication of expected number of mutations does not seem to work exceptionally well. The fact that the model is particularly doing worse in predicting mutations in melanoma and lung cancers, that are mostly affected by the exogenous factors such as UV and smoking, raises concerns that the model is too simplistic and requires a more complex version to reflect the complexity of cancers and clonal evolution in different tissues. Additionally, my worry is that these two cancers have high burden of mutations which might particularly pronounce the shortcoming of the model, and this is less visible in cancers with low mutation burden.

We thank the reviewer for this important comment. Our model did not originally account for the presence of environmental and inherited factors. The fact that LUAD and SKCM are outliers in Fig 1b is therefore a strength of the model as it shows how the endogenous process (R) by itself cannot explain cancer incidence or the number of mutations in cancers that are well-known to be affected by environmental exposures (E). To account for known exogenous factors such as UV light and tobacco smoking, thanks to the reviewer's comment, we now provide additional Figures 1c and 1d, which provide a comparison between the expected number of mutations and the observed number of mutations, under the assumption that the mutation rate has been now increased to be four times higher than the background mutation rate in the lungs and skin. This unsurprisingly reduces the deviation of the predictions from the observations, as also shown in Suppl Table S2. To further improve the impact of the paper, we have also added in the main text and in the Supplementary Information a new measure of the deviation of the expected number of mutations from the observed number of mutations for each cancer type providing a prediction for how large the effect of R versus E and H factors is (Suppl Table S2).

Another key result we have now added is that we used this new distance to estimate the background mutation rate which minimizes the mean distance across cancer types (including or not the outliers LUAD and SKCM). That estimate is equal to 0.024, which is striking as this is very close to the parameter value we originally chose for our model as provided by the available literature.

Finally, we have added in the first paragraph of the discussion the following explanation: "The distance between the lifetime cancer risk of a given tissue (a point in Fig 4) and the expected risk for that tissue (the plane in Fig 4) provides a framework for inferring the risk of each cancer subtype that may be contributed by either an E or H factor. Specifically, the larger the distance, the greater the evidence for an E or H factor."

It would be informative to know how this model holds if applied on different data other than TCGA such as PCAWG.

We thank the reviewer for this fundamental comment. As requested by the reviewer and the editor we have now extended our analyses beyond the TCGA dataset and have applied to the available sequencing data in the International Cancer Genome Consortium (ICGC) and reported its result in Fig 1d and Suppl Fig S2b. As commented also in the main text this further analysis has yielded comparable results to our results on TCGA data.

Currently, the model is based on cell replication error. Although this was thought to be the main player in cancer evolution, recent studies such as Robinson et al., Nature 2021 (<https://www.nature.com/articles/s41588-021-00930-y>) have shown that human tissues can physiologically tolerate ubiquitously elevated mutation burdens. Such burdens do not support a model in which all features of cancer and ageing are attributed to widespread cell malfunction, directly resulting in high somatic mutation burdens. Moreover, recent work by Abascal et al., Nature 2021 (<https://www.nature.com/articles/s41586-021-03477-4>) has showed that endogenous signatures attributed to ageing and cancer are also operative in post-mitotic tissues such as neurons. Hence, cell replication error cannot be the main contributor to ageing and cancers. Therefore, this raises concerns whether the current choices for this model, which are heavily weighted towards the cell replication errors in cancer, can predict a realistic cancer risk.

We thank the reviewer for these references which are important, and we now comment upon them in the discussion section of the main text. It is true that our model is based upon the fact that cells accumulate mutations during their lifespan and that this provides a major engine for tumor evolution, and we are aware of (and have now cited) the study by Robinson et al., Nature 2021 showing that replication errors may not be the main contributors to aging. At the same time, we would like to point out that the very abstract of that paper clearly states both that 1) “these results do not support a model in which all features of aging are attributable to widespread cell malfunction directly resulting from somatic mutation burdens accrued during life”, but also that 2) this applies to aging *but not to cancer risk* when it says “*Except for increased cancer risk, individuals with germline POLE/POLD1 mutations do not exhibit overt features of premature aging.*” With respect to Abascal et al., it is clear that endogenous processes are operative also in post mitotic tissues such as neurons as the reviewer pointed out. In fact, our results provide support for that, precisely in brain cancer, where we show that for glioblastoma (GBM in Suppl Fig S3) the amount of mutations accumulating exclusively at cell division are not sufficient to explain the observed mutational load. While we never claimed that endogenous processes operate exclusively at cell division, we still believe that to be the major mutational “engine” in many cases, with some important exceptions of which glioblastoma is an example.

Minor concerns

The language used to describe the model is not particularly accessible to the readers of this journal. Some part can be simplified.

Given that the mathematical details are always not particularly accessible to the general readership, we have now moved a substantial amount of the mathematical modeling to the Methods section.

Figure 1 some cancer types are indistinguishable, as they have same or very similar colours.

We have changed the color scheme in Figure 1 such that the different cancer types are now more easily distinguishable.

REVIEWER COMMENTS

Reviewer #1 (Remarks to the Author):

The three main results highlighted in the abstract are 1) the model provides a prediction for the expected number of mutations present in various tissue types at a given age and 2) how time to cancer can be approximated by a Weibull distribution and 3) predictions of lifetime cancer risk as a function of stem cells, mutation rate, and number of required mutations.

1. I'm still uncertain about the parameterization process of the model that leads to the conclusion that R factors (bad luck) can explain most of the variation in cancer risk or mutation burden. The expanded explanation of figure 1 is helpful, but it's still unclear to me whether parameters in Table 1 are chosen based on literature estimates, fit to the data, or some combination of both. It's difficult to determine the effect of the fitting process on the estimates of non-R factors, as the deviation from fits is used as evidence for non-R factors.

2. To further complicate matters, on page 5, the authors note that Figure 2 results in either good fits or undershooting. In figure 2, it appears that two disease types generally overshoot (BRCA, LUAD), while two disease types generally undershoot (LAML, COAD). Can the fitting process of those that overshoot be improved, and thus by implication reduce the model's predicted role of non-R factors?

3. It's further complicated by the fact that the authors are hesitant to use the fitting process to estimate the underlying number of driver mutations (2 or 3), rather repeating the analysis for both across the full range of diseases. Later, in figure 4, cancer-specific values for driver number, "n" are used. It's not clear if these driver values found in figure 4 can be propagated back to figure 1 and 2.

4. Response to Q1: the extra analysis provided in figure 2e is helpful and a nice modeling result, inferring background mutation rate.

5. Response to Q2: the additional discussion in the text is helpful, thank you.

6. Response to Q3: the additional figure is helpful and provides insight into disease-specific dynamics predicted by the model

In summary, my concern that this paper is more suited to a mathematical audience remains, as most of the key results are relegated to the supplemental section, but the main text is difficult to follow without referring to the supplemental text.

Minor comments:

1. Given the authors goal to introduce an analytical expression describing cancer risk / mutation burden (as noted in the introduction), I believe that the mathematical expression underlying figures 1 and 2 should be in the main text (e.g. formula 3 or 4).
2. It's difficult to assess the goodness of fit of the surface in figure 4 based on the visualization shown, as the distance of each dot in 3-dimensional space from the surface is impossible to infer by eye.

Reviewer #3 (Remarks to the Author):

The authors have mostly addressed my questions and the new version of the manuscript has improved. However, I still think the narrative of the manuscript is not accessible to the majority of readers of this journal.

Authors' Replies to Reviewers' Comments

Reviewer #1

The three main results highlighted in the abstract are 1) the model provides a prediction for the expected number of mutations present in various tissue types at a given age and 2) how time to cancer can be approximated by a Weibull distribution and 3) predictions of lifetime cancer risk as a function of stem cells, mutation rate, and number of required mutations.

1. I'm still uncertain about the parameterization process of the model that leads to the conclusion that R factors (bad luck) can explain most of the variation in cancer risk or mutation burden. The expanded explanation of figure 1 is helpful, but it's still unclear to me whether parameters in Table 1 are chosen based on literature estimates, fit to the data, or some combination of both. It's difficult to determine the effect of the fitting process on the estimates of non-R factors, as the deviation from fits is used as evidence for non-R factors.

We thank the reviewer for pointing out to a potential source of confusion that we have now clarified in the main text, where we point out that the parameters in Table 1 were not fit in any way but are chosen based on literature estimates. Therefore, the deviations from the predicted values in Figures 1 and 2 can be used as a measure of evidence for the effect of environmental exposures, given that our predictions are based on those literature's estimates rather than fitting. In this way, we don't fall into a circular logical fallacy. In the main text we now state: "We would like to point out that in these predictions there is no fitting of any of the parameters to the observed sequencing data. The number of divisions during the tissue development phase and yearly number of divisions in a healthy and in a cancer tissue in Table 1 are obtained from other studies taken from the literature. Even the Weibull distribution used in formula (4) is only fitted to epidemiological data, specifically the cancer incidence curve."

2. To further complicate matters, on page 5, the authors note that Figure 2 results in either good fits or undershooting. In figure 2, it appears that two disease types generally overshoot (BRCA, LUAD), while two disease types generally undershoot (LAML, COAD). Can the fitting process of those that overshoot be improved, and thus by implication reduce the model's predicted role of non-R factors?

We agree with the reviewer's sense that there may still be improvements to be made in the fit. The key obstacle for improving the fitting is given by the limited biological parameter estimates (in particular cell division rates) and the limits in the available sequencing data, which are cross-sectional rather than longitudinal, and often lacking proper clinical annotation. We hope that this study, which we consider a first-of-its-kind, will stimulate the field to collect better data that will improve the fitting of these types of models.

We could of course improve the fitting of the predictions by treating some of the parameters as "free parameters", which may then appear to reduce the role of environmental factors. However, that approach would be deceptive, because by fitting the data we would not be able to then attribute that good fit to the replicative processes (R factors) only. And this in turn would defeat the purpose of the study.

3. It's further complicated by the fact that the authors are hesitant to use the fitting process to estimate the underlying number of driver mutations (2 or 3), rather repeating the analysis for both across the full range of diseases. Later, in figure 4, cancer-specific values for driver

number, “n” are used. It’s not clear if these driver values found in figure 4 can be propagated back to figure 1 and 2.

As the reviewer suggested, the cancer-specific values for the number of drivers, provided in Figure 4, can indeed be propagated back. Therefore, we have now added a new analysis with the associated Supplementary Figure S3, where we provide the results using those values, and the same in Figure 2 for the selected subset of tissues.

4. Response to Q1: the extra analysis provided in figure 2e is helpful and a nice modeling result, inferring background mutation rate.

5. Response to Q2: the additional discussion in the text is helpful, thank you.

6. Response to Q3: the additional figure is helpful and provides insight into disease-specific dynamics predicted by the model

In summary, my concern that this paper is more suited to a mathematical audience remains, as most of the key results are relegated to the supplemental section, but the main text is difficult to follow without referring to the supplemental text.

We thank the reviewer for his comment that has yielded a significant rewriting of the main text, specifically we have made the reading easier for the non-mathematical audiences by removing the most complex passages that proved to be superfluous for the understanding of the main results.

We have also taken into consideration the reviewer’s comment below (see minor comment number 1) and have now added formulas (1) and (2) into the main text along with an intuitive explanation to help the reader easily follow the main text without referring to any supplemental information. Finally, we provide in the Supplementary Methods the full details of the mathematical content, thus giving the interested mathematically-oriented audience the possibility to have the full description of the model by reading the SM only, without the need to keep referring to the main article.

Minor comments:

1. Given the authors goal to introduce an analytical expression describing cancer risk / mutation burden (as noted in the introduction), I believe that the mathematical expression underlying figures 1 and 2 should be in the main text (e.g. formula 3 or 4).

We thank the reviewer for this important suggestion and have now added in the main text formulas (1) and (2) underlying Figures 1 and 2, along with a very intuitive explanation behind these formulas.

2. It’s difficult to assess the goodness of fit of the surface in figure 4 based on the visualization shown, as the distance of each dot in 3-dimensional space from the surface is impossible to infer by eye.

We have now produced a complete new figure which we feel makes it much better to visualize the position of the points with respect to the surface.

Reviewer #3

The authors have mostly addressed my questions and the new version of the manuscript has improved. However, I still think the narrative of the manuscript is not accessible to the majority of readers of this journal.

We have addressed the reviewer's concern and have made the reading significantly easier for the non-mathematical audiences. For this purpose, we have removed the most complex passages that proved to be superfluous for the understanding of the main results, and we are now providing more intuitive explanations for the few formulas present in the main text.

REVIEWERS' COMMENTS

Reviewer #1 (Remarks to the Author):

Major comments:

The author's have made significant improvements to the readability of the text & the explanation of the modeling. In particular, the changes made to the text in the caption of Table 1 improved the clarity of my understanding of the manuscript, and I now see what is derived from literature and what predictions the model computes. These changes, combined with the formulas moved from supplemental to main text generally satisfactorily address our previous main 3 concerns.

My final minor comment is that I think the authors might make a note of some potential causes of data undershoots by the model. When data overshoots the model, this provides some evidence of the importance of E & H factors. However, undershooting is fundamentally different, as E & H factors are not expected to be negative. What are potential causes of undershooting in the model (e.g. over-estimation of mutation rate)? A minor comment within the main text (in the discussion or around explanation of delta in lines 170-190) would suffice.

Minor comments:

Authors have satisfactorily address our previous 2 minor concerns. Thank you for updating figure 4 and moving some equations to the main text.

Authors' Replies to Reviewers' Comments

Reviewer #1

Major comments:

The author's have made significant improvements to the readability of the text & the explanation of the modeling. In particular, the changes made to the text in the caption of Table 1 improved the clarity of my understanding of the manuscript, and I now see what is derived from literature and what predictions the model computes. These changes, combined with the formulas moved from supplemental to main text generally satisfactorily address our previous main 3 concerns.

My final minor comment is that I think the authors might make a note of some potential causes of data undershoots by the model. When data overshoots the model, this provides some evidence of the importance of E & H factors. However, undershooting is fundamentally different, as E & H factors are not expected to be negative. What are potential causes of undershooting in the model (e.g. over-estimation of mutation rate)? A minor comment within the main text (in the discussion or around explanation of delta in lines 170-190) would suffice.

We thank the reviewer for this suggestion and have now added the following comment within the main text (p7): "Both overshooting and undershooting, if present at the population level, can be caused for example by errors in the estimates we use for the division rate, the mutation probability, or by a difference in the ability of the immune system to fight tumor cells across different organs. Additionally, when our predictions undershoot the number of mutations actually observed, whether for the whole population or for a subset of patients (points above the grey identity line in Fig. 1 or above the red line in Fig. 2), this can be due to environmental exposures (e.g., smoking or sun exposure), germline mutations (e.g., BRCA1-2 in breast cancer) or genomic instabilities (e.g., microsatellite instability in colorectal cancer) affecting these patients.

Minor comments:

Authors have satisfactorily addressed our previous 2 minor concerns. Thank you for updating figure 4 and moving some equations to the main text.